# LayerDecompose: Exploring Weight Sharing for Large Language Model Compression

## Abstract

Recent advances in large language model (LLM) compression have predominantly focused on pruning and low-rank factorization, leaving weight sharing—despite its success in classical neural network compression—largely unexplored. We introduce LayerDecompose, a novel framework that reduces parameter redundancy by sharing a core weight matrix across transformer layers and augmenting each layer with lightweight, low-rank adapters. Unlike prior SVD- and pruning-based methods, our joint optimization of shared weights and residual adapters achieves a 30% model size reduction while retaining 89% of the original performance on seven standard benchmarks. Experiments on LLaMA and other models demonstrate that LayerDecompose consistently outperforms state-of-the-art baselines. These results highlight the promise of combining weight sharing with low-rank adaptation for efficient, scalable LLM deployment. [1]

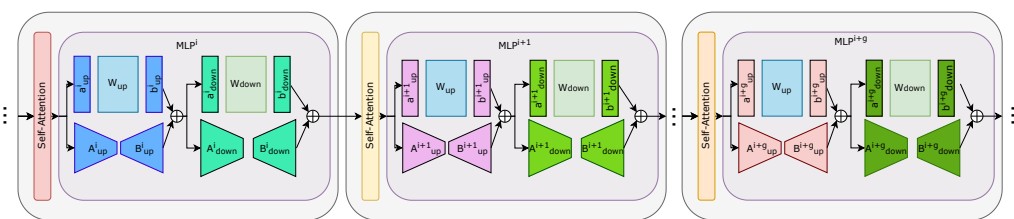

Figure 1: Schematic overview of the proposed approach. Within a group of size $g$ weights of each type (e.g. up and down projections in MLP) are shared between transformer blocks, but have a unique low-rank residuals and scaling, which are optimized to match the original weights. This decomposition is also applied to the self attention layer, omitted for brevity.

## 1 Introduction

Transformers underpin virtually every state-of-the-art large language model (LLM) today, delivering remarkable capabilities in tasks ranging from question answering and commonsense reasoning to code generation and dialogue. As model capacities have grown—from millions to hundreds of billions of parameters—the computational and memory demands for both training and inference have skyrocketed. Such scaling presents a formidable barrier to deploying these models in real-world settings, especially on resource-constrained hardware or at low latency. To bridge this gap, a rich body of work has explored post-training compression techniques—quantization, pruning, and low-rank factorization—that reduce model size and accelerate inference while striving to preserve performance.

Quantization methods Lin et al. (2024); Frantar et al. (2022) map high-precision weights to lower-bit representations, offering dramatic memory savings but often requiring hardware support for efficient low-bit arithmetic. Unstructured pruning Frantar & Alistarh (2023); Li et al. (2023) discards individual parameters based on some importance criterion, yet its resulting sparsity patterns can be difficult to exploit without specialized sparse-compute kernels. Structured pruning Zhang et al. (2024); Wei

---

[1]Code for reproducing all experiments will be released upon acceptance.

| Ratio | Method | OBQA | PIQA | HellaS. | WinoG. | ARC-e | ARC-c | MathQA | AVG | RP (%) |
|---|---|---|---|---|---|---|---|---|---|---|
| 0 % | Uncompressed | 0.44 | 0.79 | 0.76 | 0.70 | 0.73 | 0.46 | 0.27 | 0.59 | 100.0 |
| 20 % | SVD-LLM V2 | 0.32 | 0.75 | 0.52 | **0.70** | **0.72** | 0.29 | 0.24 | 0.51 | 85.2 |
| | Basis Sharing | 0.28 | 0.71 | 0.46 | 0.66 | 0.66 | 0.36 | 0.25 | 0.48 | 81.4 |
| | LLM-Pruner | 0.39 | 0.76 | 0.68 | 0.64 | 0.52 | **0.38** | 0.24 | 0.52 | 87.0 |
| | LAYERDECOMPOSE (ours) | **0.40** | **0.77** | **0.71** | 0.69 | 0.65 | 0.37 | **0.25** | **0.55** | **92.3** |
| 30 % | SVD-LLM | 0.20 | 0.65 | 0.37 | 0.59 | 0.48 | 0.26 | 0.22 | 0.40 | 66.7 |
| | Basis Sharing | 0.27 | 0.68 | 0.40 | 0.63 | **0.63** | 0.30 | 0.24 | 0.45 | 75.9 |
| | LLM-Pruner | 0.39 | 0.75 | 0.63 | 0.61 | 0.48 | 0.35 | 0.23 | 0.49 | 82.7 |
| | LAYERDECOMPOSE (ours) | **0.39** | **0.75** | **0.67** | **0.64** | 0.62 | **0.37** | **0.24** | **0.53** | **88.9** |
| 50 % | SVD-LLM | 0.16 | 0.55 | 0.27 | 0.50 | 0.28 | 0.22 | 0.21 | 0.31 | 52.7 |
| | Basis Sharing | 0.18 | 0.58 | 0.31 | 0.57 | 0.42 | 0.23 | 0.22 | 0.36 | 60.4 |
| | LLM-Pruner | **0.35** | 0.66 | 0.45 | 0.54 | 0.41 | **0.30** | **0.23** | 0.42 | 70.6 |
| | LAYERDECOMPOSE (ours) | 0.33 | **0.68** | **0.49** | **0.59** | **0.47** | 0.26 | 0.21 | **0.43** | **73.2** |

Table 1: Accuracy of *LLaMA-7B* after various compression ratios on seven benchmarks. **AVG** is the mean accuracy; **RP** is the average accuracy expressed as a percentage of the uncompressed baseline. Best compressed results are in bold.

et al. (2024) removes entire neurons or attention heads to maintain dense linear algebra, but aggressiveness can quickly degrade model quality. Low-rank adaptation approaches—exemplified by LoRA and its variants—reparameterize pretrained weights with rank-constrained updates, reducing fine-tuning cost but typically leaving the bulk of the original dense weights intact. Each of these strategies trades off ease of deployment, hardware compatibility, and final model accuracy.

In contrast to the extensive exploration of pruning and low-rank methods, weight sharing—one of the oldest and most general compression ideas in neural networks—has received surprisingly little attention for LLMs. Classic works such as the Universal Transformer Dehghani et al. (2019) and ALBERT Lan et al. (2020) have shown that sharing the same parameters across all layers can dramatically cut model size with only a modest hit to accuracy, yet naively tying weights across dozens of transformer blocks often yields unsatisfactory performance. A more nuanced form of weight sharing, combined with layer-specific lightweight adaptations, promises to balance redundancy elimination with expressive power, but has not been systematically studied in the context of large pretrained transformers.

In this paper, we introduce LAYERDECOMPOSE, a novel compression framework that leverages weight sharing across groups of transformer layers together with low-rank residual adapters and scaling to reduce parameter redundancy. Our core observation is that key transformer blocks express similar linear transformations up to permutation invariances. By learning a single shared "base" weight matrix for each group of layers and modeling inter-layer differences via trainable low-rank adapters, LAYERDECOMPOSE achieves up to 30% reduction in model size while retaining over 89% of original performance on seven standard benchmarks. Crucially, we jointly optimize both the shared weights and the residual factors in a two-stage procedure—closed-form initialization via truncated SVD followed by gradient-based refinement.

**Contributions.** Our main contributions are:

- We propose a hybrid weight-sharing and low-rank decomposition that represents a group of $m$ corresponding linear layers with a single shared matrix $W$ plus layer-specific residual factors $\{A_i B_i\}_{i=1}^{m}$ and scaling vectors $\{a_i b_i\}_{i=1}^{m}$, reducing parameters from $mn^2$ to $n^2 + 2m(nr + 1)$ with minimal extra compute.

- We characterize and exploit permutation invariances in both MLP and self-attention modules, using assignment solvers to optimally permute and align layer weights before decomposition, thereby lowering reconstruction error.

- We validate LAYERDECOMPOSE on LLaMA-7B and three additional 7B-parameter models, showing that it consistently outperforms state-of-the-art SVD- and pruning-based baselines, retaining nearly 89% of uncompressed performance at 30% size reduction across seven diverse benchmarks.

## 2 PRELIMINARIES

### 2.1 LOW-RANK ADAPTATION

LoRA Hu et al. (2022) replaces the standard linear layer $Y = XW + b$ with

$$Y = X(W + AB) + b = XW + XAB + b, \tag{1}$$

where $\text{rank}(AB) < \text{rank}(W)$. This reparameterization permits fine-tuning only the low-rank matrices $A$ and $B$, greatly reducing memory usage. Subsequent works have explored modified initializations Meng et al. (2024), alternative reparameterizations Liu et al. (2024b); Kopiczko et al. (2024); Lingam et al. (2024); Liu et al. (2024a), and revised optimization strategies Hayou et al. (2024); Zhang et al. (2023).

### 2.2 SVD-BASED MODEL COMPRESSION

Large language models require a significant amount of memory and computational power to operate. To reduce these resource demands, various model compression techniques have been developed. One approach to reducing the parameter count is to factorize the weight matrix $W \in \mathbb{R}^{m \times n}$ into a product of two matrices with fewer total parameters, $AB$, where $A, B^T \in \mathbb{R}^{m \times \tilde{n}}$ and $\tilde{n} < n$, while striving to retain as much model performance as possible. A substantial body of work applies Singular Value Decomposition (SVD) to address this problem.

An early work Winata et al. (2019) applies SVD for the LSTM cell and explores the effectiveness on different NLP tasks. FWSVD Hsu et al. (2022) utilizes Fisher information to assign importance weights to the model parameters. However, computing the Fisher information matrix involves computationally expensive gradient calculations. To mitigate these costs, ASVD Yuan et al. (2023) proposes an activation-aware decomposition method, which incorporates the distribution of activations into the weight decomposition process. In this approach, the scaling matrix is designed based on the distribution patterns observed across input activation channels. SVD-LLM Wang et al. (2025c) extends this idea further by whitening the input matrix to reduce its impact on SVD truncation, with proven guarantees of achieving an optimal theoretical truncation loss. Unlike previous works, Gao et al. (2024b) developed an approach to automatically allocate various ranks to different layers using a differential hypernetwork. SVD-LLM V2 Wang et al. (2025b) adapts this idea and truncate ranks based on the loss function value.

### 2.3 WEIGHT SHARING IN NEURAL NETWORKS

One fundamental application of weight sharing in language models is embedding weight tying, where the input and output embeddings share the same weight matrix Press & Wolf (2017); Raffel et al. (2020). Another significant aspect is weight sharing across layers in deep networks. Instead of assigning each layer its own parameters, a common set of weights is employed across multiple layers, thereby reducing redundancy and lowering the overall parameter count.

This concept was initially explored in the Universal Transformer Dehghani et al. (2019), which introduced a recurrent inductive bias into the Transformer by reusing the same layer weights at every depth. ALBERT Lan et al. (2020) further demonstrated that full weight sharing in BERT Devlin et al. (2019) results in only a minor reduction in accuracy while achieving faster training, enhanced memory efficiency, and improved regularization.

More recent work has investigated weight-sharing strategies tailored for resource-constrained environments. For example, Subformer Reid et al. (2021) and MobileLLM Liu et al. (2024c) explore various methods for sharing transformer blocks to optimize performance on mobile devices. Similarly, Residualformer Xie et al. (2023) employs LoRA reparameterization with shared base weights to train speech recognition models from scratch, whereas our focus is on compressing existing pretrained models. Basis Sharing Wang et al. (2025a) concatenates the weights of a pretrained model, applies SVD, and shares the resulting basis vectors across layers. Bae et al. (2024); Mikaelyan et al. (2025) likewise decompose each layer into a shared base plus low-rank deltas, yet LAYERDE-COMPOSE goes further by (i) adding **per-channel scaling** to curb magnitude drift, (ii) introducing a **permutation-aware alignment** that solves two LSAPs to reorder QK, VO and MLP channels, (iii) employing an **alternating closed-form initialisation** to shorten healing, and (iv) validating on models up to 13B parameters, underscoring scalability.

## 3 METHOD

### 3.1 LAYER DECOMPOSITION

Transformers consist of a stack of identical layers, each containing self-attention and MLP sub-modules. Both submodules are composed of linear transformations whose parameters are stored in weight matrices.

Figure 1 illustrates our approach. Let $G$ be a set of $m$ corresponding linear layers (for example, the "up" projections of the MLP in layers 17 through 23). For each layer $i \in G$, the original computation is

$$Y = XW_i + b_i.$$

We replace this with a scaled shared base weight $W$ plus a low-rank residual for each layer $i \in G$:

$$Y = XD_{a_i}WD_{b_i} + XA_iB_i + b_i,$$

where $D_{a_i} = \text{diag}(a_i)$ and $D_{a_i} = \text{diag}(b_i)$ are diagonal matrices used to scale rows and columns of the shared base matrix $W$ similar to Wen et al. (2020). Omitting biases for simplicity, if $W \in \mathbb{R}^{n \times n}$, $|G| = m$, $a_i, b_i \in \mathbb{R}^n$ and each $A_i, B_i^T \in \mathbb{R}^{n \times r}$ with $r < n$, then the total parameters drop from $m\,n^2$ to $n^2 + m \cdot 2(nr + 1)$, at the cost of a small extra compute for the adapters.

To initialize $W, \{A_i, B_i\}$, we minimize the Frobenius-norm reconstruction loss

$$\mathcal{L}(W, A, B) = \sum_{i \in G} \big\|W_i - (W + A_iB_i)\big\|_F = \sum_{i \in G} \big\|(W_i - W) - A_iB_i\big\|_F.$$

This loss can be viewed as seeking a rank-$r$ approximation of each difference $W_i - W$. Hence, by the Eckart–Young–Mirsky theorem Eckart & Young (1936), for a fixed base $W$ the optimal low-rank factors $(A_i, B_i)$ are given by the truncated SVD of $(W_i - W)$. Conversely, when $\{A_i, B_i\}$ are held fixed, the optimal shared weight is simply the element-wise mean

$$W = \frac{1}{|G|} \sum_{i \in G} \big(W_i - A_iB_i\big).$$

After initializing via these two closed-form updates, we initialize scaling vectors as $a_i = b_i = \mathbf{1}$ and perform a final joint refinement of $W$ and all $\{A_i, B_i, D_{a_i}, D_{b_i}\}$ using Adam Diederik (2014) by minimizing the loss

$$\mathcal{L}(W, A, B, a, b) = \sum_{i \in G} \big\|(W_i - D_{a_i}WD_{b_i}) - A_iB_i\big\|_F. \tag{2}$$

The full procedure is outlined in Algorithm 1.

### 3.2 TRANSFORMER PERMUTATION INVARIANCE

Permutation invariance in transformer modules allows multiple weight configurations to produce identical outputs by appropriately reordering intermediate dimensions.

**Multi-Layer Perceptron**    A gated transformer MLP block computes

$$y = W_d\big(\sigma(W_g\,x) \odot W_u\,x\big),$$

where $\sigma$ is applied element-wise. By permuting the intermediate hidden dimensions via an $n \times n$ permutation matrix $P$ (and its inverse $P^T$), one can rearrange the rows of $W_g$ and $W_u$ without affecting the final output. Concretely, we exploit $P^TP = I$ as follows:

$$y = W_d\,P^T\Big(\sigma\big(P\,W_g\,x\big) \odot P\,W_u\,x\Big) = W_d\,P^T\Big(P\,\sigma(W_g\,x) \odot W_u\,x\Big) = W_d\big(\sigma(W_g\,x) \odot W_u\,x\big)$$

since $P^T(P\,\sigma(W_gx)) = \sigma(W_gx)$. Hence one can absorb $P$ into the weights by defining

$$W_u' = P\,W_u, \quad W_g' = P\,W_g, \quad W_d' = W_d\,P^T,$$

yielding the same output $y$. Because there are $n!$ permutation matrices of size $n$, this gives $n!$ equivalent MLP configurations.

---

**Algorithm 1** Alternating Shared $W$ Optimization

---

**Require:** Weight group $\{W_i\}_{i \in G}$, rank $r$, alternation steps $T$, Adam steps $T_{\text{adam}}$
**Ensure:** Optimal shared weight $W$, low-rank factors $\{A_i, B_i\}_{i \in G}$, scaling vectors $\{a_i, b_i\}_{i \in G}$
   $W \leftarrow \frac{1}{m} \sum_{i \in G} W_i$                                     ▷ W initialization
   **for each** $i \in G$ **do**                                ▷ $A_i, B_i$ initialization
      $(A_i, B_i) \leftarrow \text{TruncSVD}(W_i - W, \ r)$
   **end for**
   **for** $t = 1$ **to** $T$ **do**                             ▷ Alternating optimization
      $W \leftarrow \frac{1}{m} \sum_{i \in G} (W_i - A_i B_i)$
      **for each** $i \in G$ **do**
         $(A_i, B_i) \leftarrow \text{TruncSVD}(W_i - W, \ r)$
      **end for**
   **end for**
   $a, b \leftarrow \mathbf{1}$                                         ▷ Scaling initialization
   **for** $t = 1$ **to** $T_{\text{adam}}$ **do**                         ▷ Adam Optimization
      Compute $\mathcal{L}(W, A, B, a, b)$ as in Eq. 2
      Update $W, \{A_i, B_i, a_i, b_i\}$ via Adam
   **end for**

---

**Query and Key Projections** In self-attention, the query and key projections satisfy a similar invariance: permuting their shared intermediate dimensions does not alter the attention scores. Recall

$$Q = XW_Q, \quad K = XW_K, \quad V = XW_V.$$

and

$$\text{Attn}(Q, K, V) = \text{softmax}(QK^T)\, V.$$

Inserting a permutation $P$ with $P^T P = I$ into the score computation gives

$$\text{softmax}(QK^T) = \text{softmax}\big((XW_Q)(XW_K)^T\big) = \text{softmax}\big(XW_Q\, P\, P^T\, W_K^T\, X^T\big).$$

so that defining

$$W_Q' = W_Q\, P, \quad W_K' = P^T\, W_K$$

leaves $\text{softmax}(QK^T)$ unchanged.

**Value and Output projections** Previously, we showed that permuting the dimensions of $Q$ and $K$ does not alter the attention score matrix. A similar invariance holds for the Value and subsequent Output projections.

In multi-head self-attention, for each head $i = 1, \ldots, h$ we define

$$V^{(i)} = X\, W_V^{(i)}, \quad H^{(i)} = \text{softmax}\big(Q^{(i)}(K^{(i)})^T\big)\, V^{(i)}, \tag{3}$$

where $W_V^{(i)} \in \mathbb{R}^{d \times d_v}$ is the value-projection for head $i$. We then concatenate the head outputs and apply the final output projection:

$$Y = \big[\, H^{(1)}, \ldots, H^{(h)}\, \big]\, W_O, \quad W_O \in \mathbb{R}^{(h\, d_v) \times d}.$$

Any permutation of the $h$ head-blocks and of the $d_v$ channels within each head can be absorbed into the weight matrices $\{W_V^{(i)}\}$ and $W_O$. Concretely, let

$$P_{\text{blocks}} \in \{0, 1\}^{(h\, d_v) \times (h\, d_v)}, \quad P_{\text{intra}}^{(i)} \in \{0, 1\}^{d_v \times d_v} \quad (i = 1, \ldots, h), \tag{4}$$

and form

$$P = P_{\text{blocks}} \Big(\bigoplus_{i=1}^{h} P_{\text{intra}}^{(i)}\Big), \tag{5}$$

where $\bigoplus$ denotes the block-diagonal direct sum (so the $i$th diagonal block is $P_{\text{intra}}^{(i)}$). If we collect all the per-head projections into

$$W_V = \big[\, W_V^{(1)}, \ldots, W_V^{(h)}\, \big] \in \mathbb{R}^{d \times (h\, d_v)}, \quad H = \big[\, H^{(1)}, \ldots, H^{(h)}\, \big], \tag{6}$$

then one checks

$$
\begin{aligned}
Y = H\,W_O &= \mathrm{softmax}(QK^T)\,V\,W_O \\
&= \mathrm{softmax}(QK^T)\,(XW_V\,P\,P^T)\,W_O \\
&= \mathrm{softmax}(QK^T)\,(XW_V')\,W_O'
\end{aligned}
\tag{7}
$$

with
$$
W_V' := W_V\,P, \qquad W_O' := P^T\,W_O,
$$
and $P^T P = I$ guarantees the same output. Since there are $h!$ ways to permute the head-blocks and $(d_v!)^h$ ways to permute channels within each head, the total number of distinct permutations of $(W_V, W_O)$ yielding identical outputs is $h! \times (d_v!)^h$.

### 3.2.1 Finding Optimal Permutations

We leverage these permutation symmetries to reorder layer weights so that they align more closely within each group. Formally, for two weight matrices $W_i$ and $W_j$, we seek

$$
P = \arg\min_{P \in S_n} \left\| W_i - P\,W_j \right\|_F,
$$

where $S_n$ is the set of $n \times n$ permutation matrices. Here, $P$ minimizes the difference between an anchor weight and another weight in the group.

We perform this procedure separately for three components: the MLP block, the Query–Key (QK) projections, and the Value–Output (VO) projections. Note that for QK we restrict intra-head permutations to the identity ($P_{\mathrm{intra}} = I$)—permuting channels would conflict with RoPE embeddings Su et al. (2021)—and only reorder entire heads.

**MLP block** Compute a cost matrix $D \in \mathbb{R}^{n \times n}$ whose $(i, j)$ entry is

$$
D_{ij} = \|W_u^A[i,:] - W_u^B[j,:]\|_2^2 + \|W_g^A[i,:] - W_g^B[j,:]\|_2^2 + \|W_d^A[:,i] - W_d^B[:,j]\|_2^2,
\tag{8}
$$

where $W_u$, $W_g$, and $W_d$ denote the up-projection, gate, and down-projection weight matrices. Each $D_{ij}$ aggregates via sum the squared $\ell_2$ distances between row $i$ of one layer and row $j$ of another for $W_u$ and $W_g$, plus the column differences in $W_d$. We formulate the search for the optimal permutation as a linear sum assignment problem (LSAP) Burkard & Cela (1999) and solve it with an efficient solver Crouse (2016) to obtain the optimal permutation $P$.

**QK and VO projections** Here the permutation must respect the block structure of $h$ attention heads, so channels cannot be exchanged across heads. We use a two-stage approach:

1. *Intra-head alignment:* For each pair of corresponding heads, find the best channel permutation $P_{\mathrm{intra}}^{(i)}$ by solving an LSAP on the per-head weight differences.
2. *Inter-head alignment:* Compute aggregated costs between entire heads using the intra-head-aligned weights, then solve a second LSAP to determine the head-reordering permutation $P_{\mathrm{blocks}}$.

Finally, we combine these into a block-diagonal permutation as in Eq. 5, which aligns both head order and internal channels while preserving the attention outputs.

### 3.3 Healing with Distillation

Because our weight-sharing and low-rank decomposition substantially alter the original parameters, a dedicated "healing" step is required to recover performance. Following Muralidharan et al. (2024), we apply both logit-level and hidden-state distillation Hinton et al. (2015); Sanh et al. (2019) to encourage the compressed model to mimic the teacher's behavior while reducing reliance on the specific healing dataset.

Concretely, we augment the standard language modeling loss $\mathcal{L}_{LM}$ with two distillation terms:

$$
\mathcal{L} = \mathcal{L}_{LM} + \alpha\,\mathrm{KL}\big(p\,\|\,p_{\mathrm{teacher}}\big) + \beta\,\mathrm{MSE}\big(h,\,h_{\mathrm{teacher}}\big),
\tag{9}
$$

where - $\mathrm{KL}(p\,\|\,p_{\mathrm{teacher}})$ is the Kullback–Leibler divergence between the student's output distribution $p$ and the teacher's distribution $p_{\mathrm{teacher}}$, $\mathrm{MSE}(h,\,h_{\mathrm{teacher}})$ is the mean squared error between their hidden-state activations, and $\alpha, \beta$ weight these distillation terms relative to $\mathcal{L}_{LM}$.

# 4 EXPERIMENTS

## 4.1 HOW TO CHOOSE GROUPS?

To apply our decomposition, we must partition the $L$ transformer layers into groups that will share the same base weight matrix. A natural baseline is to form consecutive groups of fixed size, but ideally we would group layers whose weights are most alike so as to minimize the reconstruction loss in Eq. 2.

We first measured pairwise Frobenius distances

$$d(L_i, L_j) = \|W_i - W_j\|_F$$

between corresponding weight matrices across layers. Figure 2 shows both the distance matrix and its histogram for the MLP up-projection weights of LLaMA. The heat-map reveals no clear block structure, and the histogram is tightly centered around its mean, indicating that all layers are roughly equally dissimilar.

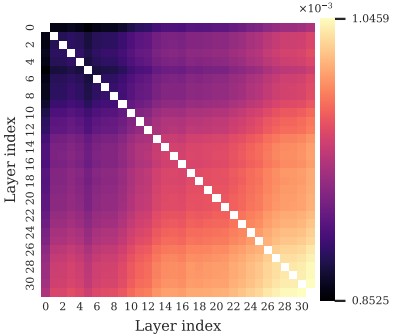 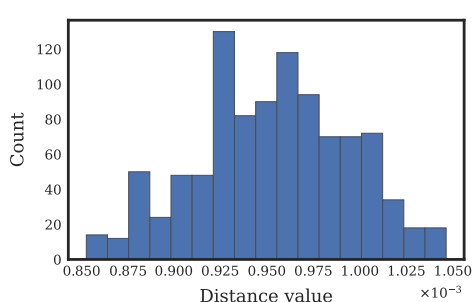

Figure 2: Pairwise Frobenius distance matrix and histogram for the MLP up-projection weights in LLaMA. The lack of visible structure and the narrow distribution of distances suggest that fixed-size consecutive grouping is as reasonable as any clustering based on these metrics.

Given the limitations of a purely weight-space metric, we next define a functional similarity measure:

$$\rho(L_i, L_j) = d\big(L_i(X_i), L_j(X_i)\big), \tag{10}$$

where $L_i$ and $L_j$ denote the $i$th and $j$th layers, $X_i$ is the actual input to $L_i$ collected during a forward pass, and $d(x, y) = \|x - y\|_2^2$. This quantity captures how closely layer $j$ can mimic layer $i$ on its native inputs. Note that $\rho$ is not symmetric in general.

Figure 3 shows the resulting similarity matrix for the MLP blocks in LLaMA. We observe a banded structure along the diagonal, indicating that adjacent layers produce more similar outputs. Furthermore, the first half of the network exhibits larger approximation errors than the second half. Motivated by these observations and for implementation simplicity, we form groups of consecutive layers for weight sharing in this work.

## 4.2 OPTIMAL PARAMETER BUDGET ALLOCATION

Even with consecutive grouping, our framework has three key hyperparameters under a fixed parameter budget: the span of affected layers, the size of each group, and the adapter rank. With a fixed parameter budget, one can either apply strong compression to a few layers or perform milder compression across a larger number of layers.

To identify which regions of the network tolerate compression best, we first compressed a single block of ten consecutive layers at a time. Figure 4 shows that compressing the earliest or latest layers incurs large perplexity increases, whereas targeting the final third of layers yields the smallest degradation. These findings corroborate prior analyses of layer sensitivity Gromov et al. (2025); Men et al. (2024); Wang et al. (2024).

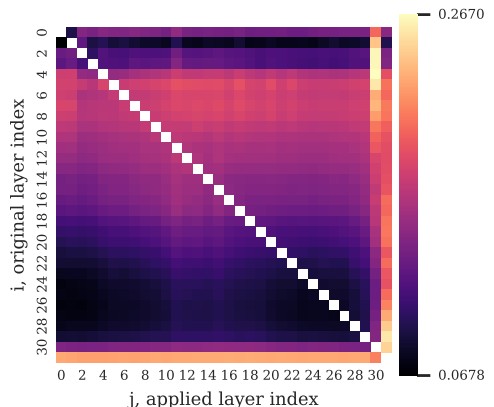

Figure 3: Functional similarity matrix for MLP blocks in the LLaMA model, where each entry $(i, j)$ is given by $\rho(L_i, L_j)$. Lower values along the diagonal indicate that nearby layers are more functionally similar.

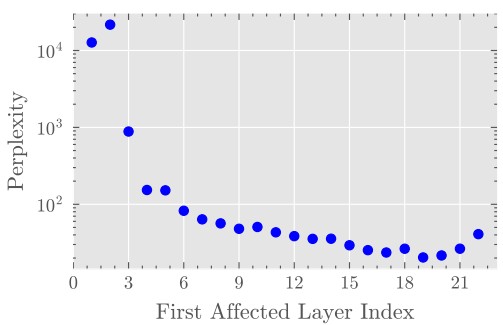

Figure 4: Perplexity after decomposition (before healing) for a single group of 10 layers. Each x-value denotes the index of the first layer in the compressed block (e.g., 19 covers layers 19–29).

We then performed an exhaustive search over hyperparameters under a fixed budget. For a 30% compression of LLaMA-7B (4.7 B parameters), we fixed the final layer at index 30 and varied the starting layer and group size. Given each choice of start and group, we computed the adapter rank to exactly match the remaining parameter budget (see appendix B for more details). One clear trend is that applying milder compression over a wider range of layers—using larger residual ranks—yields better perplexity than more aggressive compression on a smaller subset of layers.

### 4.3 MAIN RESULTS

We first evaluate the effectiveness of LAYERDECOMPOSE on *LLaMA-7B* Touvron et al. (2023a). Compression ratios from 20–50 % are compared against three state-of-the-art baselines. **SVD-LLM** (Wang et al., 2025c;b) compresses weights via singular-value decomposition combined with a whitening transform and is the strongest published SVD variant to date. **LLM-Pruner** (Ma et al., 2023) performs gradient-based structured pruning, while **Basis Sharing** (Wang et al., 2025a) concatenates layer weights, applies SVD to the joint matrix, and shares the resulting basis vectors across the group. All baselines, like our method, apply a post-compression healing phase; hyper-parameters and further details appear in Appendix A.

We retain the original evaluation protocol of *LM-Evaluation-Harness* Gao et al. (2024a) and report accuracy on seven benchmarks covering question-answering and commonsense reasoning: OpenBookQA (OBQA) Mihaylov et al. (2018), PIQA Bisk et al. (2020), HellaSwag Zellers et al. (2019), WinoGrande Sakaguchi et al. (2019), ARC-Easy and ARC-Challenge Clark et al. (2018), and MathQA Valentino et al. (2024). In addition to absolute accuracy, we compute *Relative Performance* (RP), the ratio of a compressed model's average accuracy to that of the uncompressed model.

Table 1 demonstrates that LAYERDECOMPOSE achieves the highest average accuracy and relative performance, retaining approximately 89% of the uncompressed model's quality while matching or surpassing each baseline. These findings underscore the effectiveness of weight sharing in compressing large language models.

To test generality, we apply the same 30 % compression to three other 7B-parameter models—*Qwen-7B* (Bai et al., 2023), *DeepSeek-7B* (DeepSeek-AI et al., 2024), and *OLMo-7B* (Groeneveld et al., 2024). Results appear in Table 2.

To further assess scalability, we compress the 13B-parameter *LLaMA-2* model (Touvron et al., 2023b). Table 3 shows that LAYERDECOMPOSE retains 94.6 % of baseline performance at 20 % compression and 89.6 % at 30 %. The higher RP compared with the 7B case indicates that larger

| Model | AVG | RP (%) |
|---|---|---|
| Qwen-7B | 0.50 | 83.0 |
| DeepSeek-7B | 0.52 | 88.2 |
| OLMo-7B | 0.48 | 84.0 |

Table 2: Average accuracy (AVG) and relative performance (RP) of LAYERDECOMPOSE on additional 7B models under 30% compression.

| Compression | AVG | RP (%) |
|---|---|---|
| 0 % | 0.62 | 100.0 |
| 20 % | 0.59 | 94.6 |
| 30 % | 0.56 | 89.6 |
| 50 % | 0.42 | 67.3 |

Table 3: Average accuracy (AVG) and relative performance (RP) of LAYERDECOMPOSE on *LLaMA-2-13B* at various compression ratios.

models contain more redundancy and therefore benefit even more from our shared-adapter decomposition.

### 4.4 ABLATION STUDIES

**Effect of scaling and permutations.** To reduce reconstruction error, we augment the shared–adapter decomposition with (i) per-channel scaling vectors and (ii) permutations that align weight dimensions before factorization. Figure 5 shows that each modification yields a consistent improvement: introducing scaling lowers the loss across all tested ranks, and adding pre-decomposition permutations yields the lowest reconstruction error overall.

**Effect of distillation during healing.** We next evaluate how knowledge distillation influences the healing phase that follows decomposition. Table 4 reports perplexity on a held-out C4 subset *(i)* immediately after decomposition, *(ii)* after language-model (LM) fine-tuning alone, and *(iii)* after LM fine-tuning augmented with logit- and hidden-state distillation. Distillation provides an additional perplexity reduction of approximately 4%, confirming its value for recovering performance.

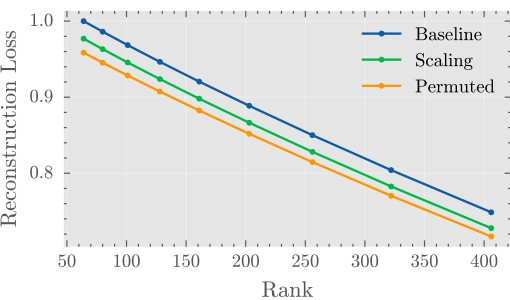

Figure 5: Normalized reconstruction loss as a function of adapter rank for three configurations: the vanilla decomposition (**Baseline**), decomposition with scaling vectors (**Scaling**), and decomposition with both scaling and pre-decomposition permutations (**Permuted**). Lower is better.

| Stage | Perplexity |
|---|---|
| After decomposition | 21.66 |
| Healed *w/o* distill. | 10.50 |
| Healed *w/* distill. | 10.08 |

Table 4: Perplexity on C4 at three stages of healing. Distillation consistently improves the healed model.

### 5 CONCLUSION

We introduced LAYERDECOMPOSE, a compression framework that represents blocks of consecutive transformer weights with a single shared matrix plus lightweight, layer-specific adapters and scaling vectors. By formalizing permutation invariances in both MLP and self-attention components, we revealed a vast family of equivalent weight configurations and leveraged these symmetries to further reduce redundancy. Empirical results on LLaMA-7B and three additional 7B and 13B-parameter models show that our weight-sharing approach matches or exceeds state-of-the-art SVD- and pruning-based compression baselines across diverse benchmarks. We believe that our findings will inspire further exploration of weight sharing as a systematic strategy for efficient LLM compression and scaling.

## REPRODUCIBILITY STATEMENT

We have described our proposed algorithm in detail in Section 3, including closed-form initialization, alternating optimization, and the healing procedure with distillation. Key equations and Algorithm 1 provide the necessary steps for reproducing our method. Hyperparameter settings, training details, and datasets are specified in Section 4 and Appendix A, with additional derivations and rank computation formulas in Appendix B. Figures 2–7 visualize grouping heuristics, sensitivity analyses, and ablation studies to support reproducibility. Code for implementing LAYERDECOMPOSE, along with scripts to replicate all experiments, will be released upon acceptance.

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

## A HEALING DETAILS AND HYPERPARAMETERS

An exhaustive search over compression configurations for the LLaMA-7B model on a subset of the development dataset revealed that the optimal perplexity is achieved using the groups `[[10, 11, 12, 13], [14, 15, 16, 17], [18, 19, 20, 21], [22, 23, 24, 25], [26, 27, 28, 29]]` with a residual rank of $r = 649$.

We apply healing on the C4 train corpus Raffel et al. (2020) for 100,000 iterations with an effective batch size of 8, truncating all sequences to a maximum length of 1,024 tokens. The weights for the distillation loss (Eq. 9) are set to $\alpha = 0.05$ and $\beta = 0.2$.

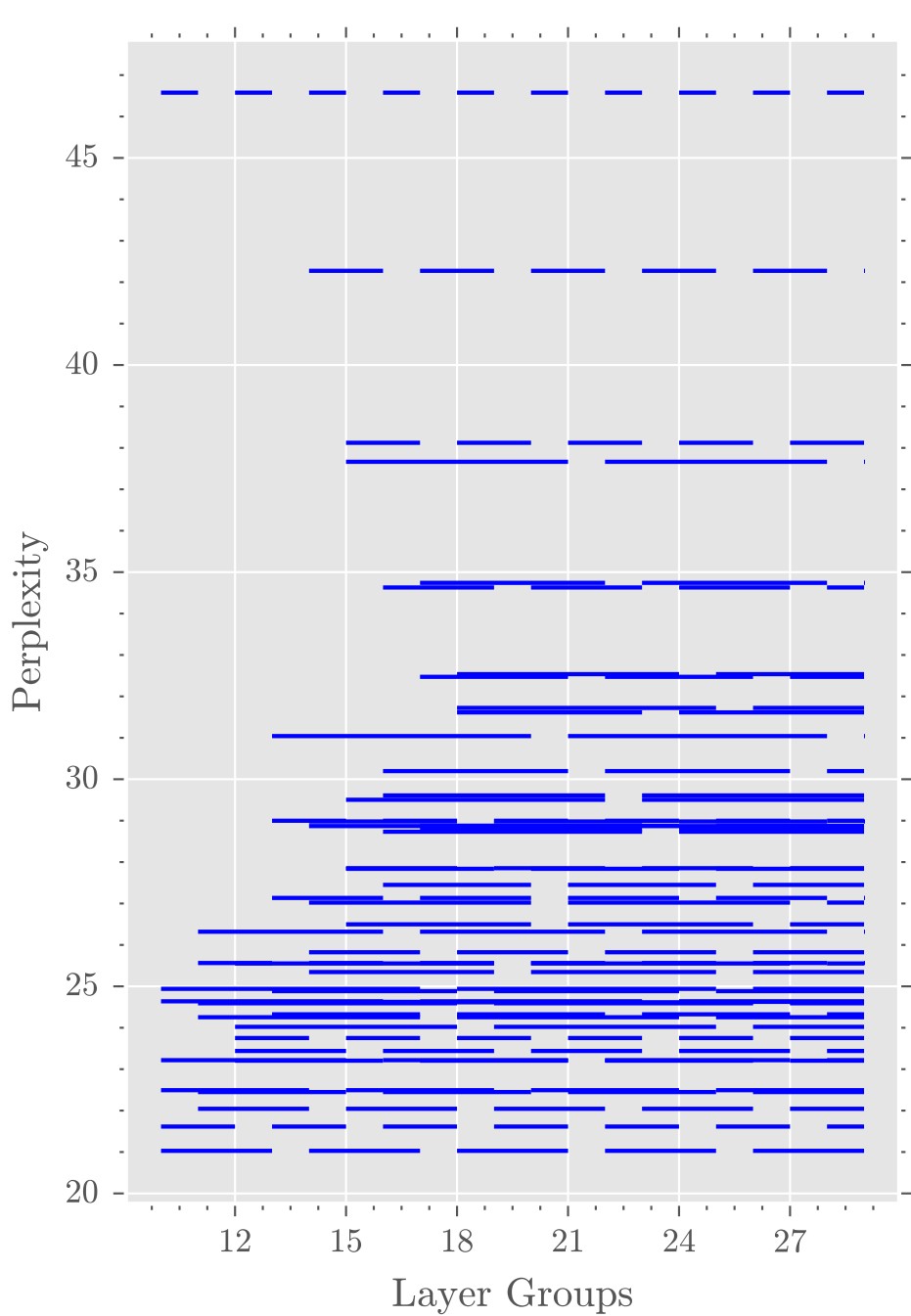

Figure 6: Perplexity after decomposition for various compression configurations. Each line corresponds to a different group of layers. Slight compression over a broader layer span—with higher adapter rank—yields better perplexity than aggressive compression on a smaller subset.

For optimisation we use Adam Diederik (2014) with learning rate `5e-5`, cosine annealing schedule with 500 warmup steps and weight decay 0.01.

Experiments were conducted using 2 NVIDIA A100 GPUs and took approximately 14 hours including evaluation for the 7B models.

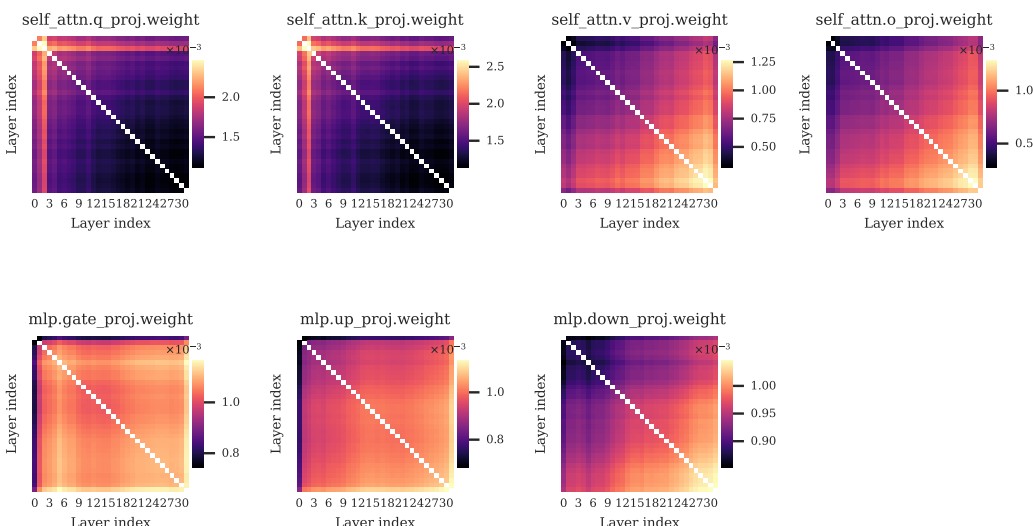

Figure 7: Heat-maps of pair-wise distances for each of the 7 layer groups. Each subplot shows the distance matrix for one layer type.

## B    RANK COMPUTATION

During the exhaustive search with fixed hyperparameters, the adapter rank could be computed in a single way as follows:

$$r = \frac{P_B - \Big(P_{nb} + (L_T - L_A) \cdot P_l + G \cdot P_l\Big)}{L_A \cdot (d_I + d_O)},$$  (11)

where:

- $r$ is the adapter rank,
- $P_B$ is the fixed total parameter budget,
- $P_{nb}$ denotes the number of parameters that are not subject to compression (e.g. embedding and LM head layers),
- $L_T$ is the total number of layers in the model,
- $L_A$ is the number of affected layers, i.e. selected for compression,
- $P_l$ represents the number of parameters in one layer,
- $G$ is the number of groups within the affected layers,
- $d_I$ and $d_O$ are the sum of input and output dimensions of a layer, respectively.

Figure 6 presents the results of this exhaustive search. The performance varies substantially across configurations. One clear trend is that applying milder compression over a wider range of layers—using larger residual ranks—yields better perplexity than more aggressive compression on a smaller subset of layers.

## C    EXTRA WEIGHT DISTANCES

Pairwise Frobenius distances for all layer types in a transformer(LLaMA-7B) are depicted on the Fig. 7. Later layers tend to be less similar.

## D    LLM USAGE

Large language models were used only for minor editorial polishing of text and code autocompletion during implementation. They did not contribute to research ideation, experimental design, or substantive writing.

