# OpenReview forum: "LayerDecompose: Exploring weight sharing for Large Language Model Compression"
_ICLR.cc/2026/Conference — Submitted to ICLR 2026_

### Official Review · Reviewer_jPAH · 2025-10-27

**Soundness:** 2
**Presentation:** 1
**Contribution:** 2
**Rating:** 4
**Confidence:** 4

**Summary:**

This paper introduces LayerDecompose, a compression framework for large language models that combines weight sharing with low-rank adapters. The key idea is to represent groups of consecutive layers with a single shared weight matrix W, augmented with layer-specific low-rank residuals and per-channel scaling vectors. The authors exploit permutation invariances in transformer modules to better align weights before decomposition. Experiments on LLaMA-7B and other models show 30% compression while retaining 89% of original performance.

**Strengths:**

1. Clear methodology: The paper provides a well-structured algorithmic approach (Algorithm 1) with closed-form initialization via alternating SVD followed by joint optimization.
2. Comprehensive experiments: Evaluation across multiple models (LLaMA-7B, Qwen-7B, DeepSeek-7B, OLMo-7B, LLaMA-2-13B) and seven benchmarks demonstrates generalizability.
3. Comparison against relevant baselines (SVD-LLM, LLM-Pruner, Basis Sharing).

**Weaknesses:**

1. Missing results. Table 1 only shows results after healing/distillation. No comparison of pre-healing performance across methods. Cannot assess whether improvements come from the decomposition itself or the healing procedure. Please add a column showing "After Decomposition (Before Healing)" for all methods.
2. Inconsistent baseline results. LLM-Pruner results in Table 1 differ significantly from the original paper. Example: At 50% compression, paper reports PIQA=0.66, HellaSwag=0.45, ARC-e=0.41, while original LLM-Pruner reports PIQA=0.693, HellaSwag=0.476, ARC-e=0.465. This raises concerns about the experimental setup and fair comparison. Please clarify the experimental protocol, verify baseline implementation, or cite reasons for differences.
3. Limited novelty over prior work. DeltaLLM (Mikaelyan et al. 2025) already propose shared base + low-rank deltas. While the paper mentions differences (scaling, permutation-aware alignment, closed-form init), the core contribution feels incremental. The permutation alignment is the main novel component, but its impact is not isolated.
4. Unclear visualization (Figure 3): Authors claim "banded structure along diagonal" showing adjacent layers are similar. But the pattern is not visually obvious; quantitative analysis would help
5. SVD dimension error (Line 124) States A, B^T ∈ ℝ^{m×ñ} but for standard SVD, shapes should be A ∈ ℝ^{m×r}, B ∈ ℝ^{n×r}

**Questions:**

see weakness

---

### Official Review · Reviewer_1SsX · 2025-10-30

**Soundness:** 3
**Presentation:** 3
**Contribution:** 2
**Rating:** 4
**Confidence:** 5

**Summary:**

This paper introduces LAYERDECOMPOSE, an LLM compression approach leveraging weight sharing. The method compresses models by sharing a single "base" weight matrix across a group of transformer layers while augmenting each layer with lightweight, low-rank residual adapters and scaling vectors. Notably, it permutes the weights before decomposition, minimizing reconstruction error. The authors demonstrate that this approach can achieve a 30% reduction in model size on LLaMA-7B while retaining 88.9% of its original performance on seven benchmarks, outperforming SVD- and pruning-based baselines.

**Strengths:**

1. The paper is well-written and easy to follow. It tackles the critical problem of weight compression in LLMs.
2. The idea of applying permutation alignment to weights to maximize the similarity of the base matrix is insightful.

**Weaknesses:**

1. The models used in the experiments are outdated. Specifically, earlier-generation LLMs like LLaMA-7B are known to be more robust to compression, whereas modern, heavily overtrained LLMs like LLaMA 3.1 or Qwen3 are much harder to compress. The work's practicality would be significantly enhanced by including experiments on these modern models.
2. The baselines are outdated. For example, modern SVD-based methods like [1], claiming performance on par with quantization at high compression ratios, are not included as baselines or cited for discussion.
3. It is unclear if the method can scale to larger models (e.g., 70B). Solving the required LSAPs could be prohibitively costly at that scale.

[1] BitStack: Any-Size Compression of Large Language Models in Variable Memory Environments

**Questions:**

1. Can you quantify the computational cost of solving the LSAPs?
2. Can you add a few experiments on modern models and baselines, or at least add a discussion about this?

---

### Official Review · Reviewer_cPRP · 2025-10-30

**Soundness:** 3
**Presentation:** 2
**Contribution:** 2
**Rating:** 2
**Confidence:** 4

**Summary:**

This paper proposes a method for compressing LLMs by sharing a core weight matrix across layers, while employing low-rank adapters for each type of weight matrix within a layer. To further reduce approximation error, the method groups output similar layers and permutes the weight matrices within each group.

**Strengths:**

1.	Same type of weight matrix in different layers shares a same matrix is interesting.
2.	More redundancy in LLMs is exploited through layer grouping and weight matrix permutation, resulting in better performance than previous methods.

**Weaknesses:**

1.	Lack of generality. All models in the experiments are implemented using MHA, whereas GQA has now become widely adopted.
2.	Unfair comparison. In prior works such as SVD-LLM and Basis-Sharing, models were calibrated using only 256 samples without any additional training, whereas this work undergoes a full distillation-based training process.
3.	No end-to-end latency reported. The additional adapters for weight matrices may introduce higher latency, but the overall impact on inference speed remains unreported.

**Questions:**

1.	In table 1, why the proposed method is compared with SVD-LLM V2 under 20% compression ratio while compared with SVD-LLM under the rest compression ratio?
2.	Which dataset is used for distillation? And how long does this distillation take?
3.	Which layers are grouped together?
4.  What is the comparison with  SVD-LLM V2 and Basis-Sharing?
[1] SVD-LLM V2: Optimizing Singular Value Truncation for Large Language Model Compression, NAACL 2025
[2] Basis Sharing: Cross-layer Parameter Sharing for Large Language Model Compression, ICLR 2025

---

### Official Review · Reviewer_ULsj · 2025-10-31

**Soundness:** 2
**Presentation:** 3
**Contribution:** 1
**Rating:** 2
**Confidence:** 4

**Summary:**

This paper proposes a novel method for model compression, exploring weight sharing between layers and using low-rank factors to compensate for errors. Its accuracy is validated on several 7B models.

**Strengths:**

1. It proposes a compression paradigm that differs from standard quantization and SVD.

2. The figures are clear and well-illustrated.

**Weaknesses:**

1. In my opinion, the paper's most significant weakness is its inability to translate storage savings into inference acceleration. Although weights are shared between layers, each layer must still load these weights during computation at inference time, meaning memory access is not reduced. Furthermore, the authors introduce a LoRa branch, which adds latency. Even with kernel fusion (e.g., as in SVDQuant), this latency is likely non-trivial. In contrast, recent mainstream SVD methods reduce parameters while offering straightforward acceleration by directly cutting the computational load.

2. The experimental section is significantly below the acceptance standard. The authors only test on outdated models and only at the 7B scale. Comparisons to other works are also limited. Comprehensive benchmarks on the Llama 2 and Llama 3 series (from 7B to 70B) are necessary to demonstrate the method's effectiveness and generalizability.

3. The proposed method requires an additional training/fine-tuning step to recover performance. How does it compare to other Post-Training Quantization (PTQ) methods when this extra training step is omitted?

In summary, this compression form seemingly increases inference latency, and the authors have failed to demonstrate its effectiveness and generalizability on a wide range of modern models. Therefore, compared to the mature, existing paradigms of quantization and SVD, I find the contribution of this new paradigm to be minimal.

**Questions:**

1. Can the authors provide experimental results on more recent models, such as Llama 3 8B and Llama 3 70B?

2. How does the method perform compared to other PTQ methods without the fine-tuning/retraining step?

3. Can the authos provide experimental results on inference speed?

---

### Meta-Review · Area_Chair_pXPr · 2025-12-30

**Summary:**

The core idea of this paper is to share weights across different layers and use low-rank components to compensate for compression error. Reviewers generally agree that the paper is clearly written and explores a compression direction that differs from standard pruning, SVD, and quantization methods. However, reviewers also identified several substantial weaknesses, including insufficient experiments on modern and large-scale models, unclear benefits for inference efficiency, and limited novelty relative to existing shared-basis and low-rank decomposition approaches.

The authors did not provide a rebuttal. Based on these considerations, the area chair finds that the identified weaknesses outweigh the strengths of the submission and therefore recommends rejection.

**Reviewer Concerns:**

1. Multiple reviewers note that it is unclear how the reduced model size translates into improved inference efficiency, and no inference latency or throughput results are provided.

2. The experimental evaluation is considered insufficient in two respects: (a) it does not include more recent models such as LLaMA-3 or Qwen-3, and (b) it does not demonstrate scalability to larger models (e.g., 70B).

3. The comparison with prior methods may not be fully fair, as many baseline approaches do not require an additional post-training or distillation stage.

4. The novelty of the proposed method relative to prior work, such as Basis Sharing and DeltaLLM, requires further clarification.

**Reviewer Scores:**

No rebuttal was submitted, and reviewer scores are assumed to remain unchanged.

---

### Decision · Program_Chairs · 2026-01-26

Reject